# A Segmentation Model of ECU Excitation Signal Based on Characteristic Parameters

**DOI:** 10.3390/s21124165

**Published:** 2021-06-17

**Authors:** Xingjian Zheng, Bo Wang, Yongqi Ge

**Affiliations:** School of Information Engineering, Ningxia University, Yinchuan 750021, China; zhengxj@nxu.edu.cn (X.Z.); readywang@nxu.edu.cn (B.W.)

**Keywords:** characteristic parameters, segmentation model, excitation signal, ECU

## Abstract

According to the basic structure and working principle of the excitation signal sensors of a diesel engine electronic control unit (ECU), a segmentation model of an ECU excitation signal based on characteristic parameters (ESCP-SM) is proposed. In the ESCP-SM, the ECU excitation signal is divided into several parts, and each part has its characteristic parameters model. By using the same global parameters and strictly controlling each part’s proportional parameters, the ESCP-SM can achieve signal alignment and dynamic frequency modulation. Based on the simulation experiment, spectrum analysis proves that this modeling method ensures that the original signal’s effective information is not lost. Pearson similarity analysis shows that the similarity between the simulation signal and practical signal reaches 74%, exhibiting strong correlation. In addition, we set up a physical testing environment. ESCP-SM is realized based on virtual instrument technology, and provides excitation signals for a Komatsu 8 ECU. By modifying the parameter configuration, the ECU can drive the injector to work correctly.

## 1. Introduction

Infrastructure construction plays an essential role in the development of a regional economy. The most often utilized equipment in infrastructure building is engineering vehicles [1]. Diesel engines are mostly employed in construction trucks. The engine ECU is the brain that controls the coordinated operation of all parts. A malfunctioning ECU might result in construction delays and accidents [2]. In the construction vehicle industry, ECU testing technology has been extensively used [3]. ECU drives the engine based on various signals from the actual sensors. The signal generator can be used in place of the actual sensor to complete dynamic measurements and engine ECU testing in an offline condition [4,5]. Therefore, the design of the signal generator is the key aspect of the dynamic testing of the engine ECU [6,7]. Signal generator testing can reduce cost and complexity by up to 90% compared to traditional testing [8]. Therefore, the accuracy and synchronization of crankshaft and camshaft signal simulation are critical to ECU offline testing.

The ECU obtains real-time engine status during engine operation by collecting various excitation signals via the sensor, including the crankshaft position (CKP) signal, the camshaft position (CMP) signal, the knock signal, and the throttle signal [9,10]. Among all excitation signals, the CKP and CMP signals are the most critical for fuel injection regulation [11,12,13]. The CKP signal and the CMP signal are a set of synchronous signals; the crankshaft turns two times, and the camshaft turns once. There are usually several missing teeth on the signal panel of the crankshaft to distinguish the number of crankshaft turns. In addition to the gear shaping, the camshaft gear slots are equally spaced. The ECU collects engine speed and cylinder information through the crankshaft position sensor (CKPS) and camshaft position sensor (CMPS) [14]. The ECU uses the unique crankshaft angle and the combined camshaft signal to identify which cylinder is operating and to precisely manage the injection time. The working principle of the CKP signal and CMP signal is shown in Figure 1.

The research on ECU excitation signal simulation is primarily focused on two factors: Hardware circuit simulation [15] and the virtual instrument signal generator. In terms of hardware circuit simulation, the signal generator is designed by a single chip microcomputer and digital to analogue conversion technology [16]. In the case of the HBS-6 type excavator ECU detector (HBS-EED), the disadvantages of hardware circuit simulation include limited scalability, low signal accuracy, and a complex circuit structure. In terms of the virtual instrument signal generator, the signal generator is designed using virtual instrument technology such as LabVIEW and MATLAB [17,18]. The precision of the analogue signal is high, and the complexity of the signal generator is reduced. However, due to the different structure of the engine, the CKP and CMP signals are numerous. A single software module cannot easily accommodate the requirements of several signals.

This paper proposes a segmentation model based on characteristic parameters for easily and quickly simulating various CKP and CMP signals. In contrast to the prior method, which concentrated only on the precision of the analogue signal, we sought to improve the signal’s variety while maintaining its accuracy and synchronization. To pursue a higher signal accuracy, we adopted the following design. Firstly, the characteristic parameters of the CKP and CMP signals were analyzed so that the signals could be modeled piecewise. Subsequently, the proportional coefficient was set to strictly control the periodic relationship of each piecewise function. Finally, the analogue signal was output through the high precision synchronous data acquisition (DAQ) card. The DAQ card is only responsible for the digital-to-analogue conversion to output analogue signals, so the model’s design can ensure the high precision of the signals. To ensure the synchronization of the CKP and CMP signals, we extracted and set the frequency of the signals. The characteristic parameters control the relative position between the missing teeth of the CKP signal and the gear shaping the CMP signal. To increase the diversity of signals, we simulated different crankshaft and camshaft signals by setting the characteristic parameters of the signals. The experimental results show that the similarity between the simulated ESCP-SM signal and the original signal is up to 74%, and the accuracy is improved by 26% compared with the hardware HBS-EED signal. This study makes the following contributions:The characteristic parameters of crankshaft position sensor (CKP) signals and camshaft position sensor (CMP) signals were analyzed. The signal type can be distinguished by detecting the threshold value of the difference between adjacent points of the signal. Fast Fourier transform (FFT) was used to obtain the frequency parameters of the original signal;A segmentation model of the ECU excitation signal based on characteristic parameters (ESCP-SM) is proposed. By the mathematical modeling of crankshaft and camshaft signals, high-precision analog signals were generated. Different CKP and CMP signals were generated by setting the characteristic parameters of the signals;The design of the ESCP-SM ensures a high signal accuracy and significantly increases signal diversity. We released the source code so that other researchers can better understand the control characteristics and operational parameters of the ECU. It provides a large quantity of valuable data for the research and evaluation of ECU simulation software. The LabVIEW source code used in this study can be found on GitHub [19].

The rest of the paper is organized as follows. In Section 2, the related work is summarized. In Section 3, we analyze the characteristic parameters of CKP and CMP signals. In Section 4, the modeling process of CKP and CMP signals is summarized, and the signal is simulated by the mathematical modeling method. In Section 5, the effectiveness of the proposed method is verified by experiments, and the the results of the method are analyzed by Pearson similarity. Finally, we conclude this work and provide some directions for future work in Section 6.

## 2. Related Work

In recent years, hardware-in-the-loop simulation (HIL) technology has been widely used in engine ECU testing [20]. The signal generator design is the essential step in the HIL process. The ECU provides for the precise control of the actuator through a series of sensors to obtain real-time operating data of the engine. The CKP and CMP signals are the most crucial input signals in the engine control system, as they are required for fuel injection and ignition timing regulation. In the relevant literature, the existing signal generation is mainly carried out considering two aspects: Single chip microcomputer (SCM) hardware simulation and virtual instrument software simulation.

In the traditional engine signal simulation platform, multiple signal generators are combined, and the signal is changed by manual adjustment. The operation is complicated, and the actual working condition cannot be simulated accurately. For example, the motor drives the optical hole disk to produce the crankshaft signal. The signal type is fixed and cannot be applied to the simulation of other kinds of engines. As the signal’s speed change can only be regulated slowly by hand and cannot accurately imitate a failure condition such as the engine speed being out of control, this solution has difficulty meeting the requirements for real-time control. To solve the problems existing in the above traditional methods, A. Z. Jidin et al. [21,22] used SCM PIC18F458 as the core to simulate the actual crankshaft signal. A stable analog voltage value is generated by pulse width modulation (PWM) and sent to the integrated voltage-controlled oscillation device CD4046. The analog signal is created at low and medium frequencies to approximate a real CKP signal, but its accuracy and adjustability are limited, its dynamic response is sluggish, and parameter adjustment is cumbersome. To address the signal frequency modulation and precision, Yao Hongqiang et al. [23] and Zuo Wenlin et al. [24] adopted FPGA as a digital platform to design PWM signal generators. The module in Quartus II6.0 is used in FPGA to make the output PWM wave frequency constant using the socket method. However, the circuit design is complex, and the signal frequency is low and difficult to debug. To simplify the circuit and improve the resolution of frequency, Jiang Pengyu et al. [25] applied C8051F MCU to design the signal generator. They encapsulated a particular square wave function and sine function to improve the precision of the signal. Generally, the signal replicated by the single hardware circuit above has complex parameter tweaking and a limited number of signal kinds.

To solve the problem that parameters are difficult to adjust, I. J. Oleagordia Aguirre et al. [26] used a digital programmable signal generator (DPSG) to generate complex and diverse signals. The signal parameters set in the software module are sent to the hardware module to adjust the signal generated by the hardware circuit. However, much work is required during the software design stage, and the development time is rather significant. To reduce the complexity of software module development, Senol Gulgonul et al. [27] established the CAM and crank signal models in the Simulink environment. The model is compiled with a Simulink encoder and flashed into the target microcontroller NXP’s MPC5744P development suite, which enables the model to be compiled and embedded into a variety of target microcontrollers. The above method of setting hardware circuits to generate signal parameters through software generally has low signal precision and complex hardware circuits.

In order to simplify the complexity of the circuit and improve the accuracy of the signal, Zhao Futang et al. [28] selected the sinusoidal signal generating function of LabVIEW and superposition of white noise to simulate the magnetoelectric crankshaft signal. The D/A conversion card of the NI series is used to output the analog signal simulated by the software to realize the signal simulation of the core sensor’s engine ECU. However, the accuracy of the simulated signal is low in the missing part of the crankshaft. To improve the signal accuracy of the simulation of crankshaft tooth absence, Lian Huijuan et al. [29] established the simulation model for the crankshaft and camshaft sensors using Matlab/Simulink/Stateflow and the arcsine function (MSS-AS). According to the different number of teeth, a sinusoidal function and arcsine function are used to fit the crankshaft signal curve, especially in the missing teeth. However, this method involves heavy programming requirements. To reduce the workload of programming, Yanqiu Zhao et al. [30] realized the seamless connection between LabVIEW and MATLAB by using the LabVIEW Simulation Interface Toolkit. Through a dynamic call to the Simulink model, an accurate simulation of the engine excitation signal is realized.

The hardware-in-the-loop simulation links a portion of the real system equipment to the computer, simulates a portion of the real system equipment using the mathematical model, and simultaneously assures the function of the whole system. Krzysztof Wiecławski et al. [31,32] introduced a model of an injector in the form of a current waveform that describes the changes in current and voltage during its operation. The actual start time and duration of injection can be determined using a mathematical model that describes the voltage-current phenomenon during the injector operation. Zhai et al. [33] analyzed a framework of the speech emotion recognition system and summarized and modeled the characteristic parameters of different speech signals. We refer to the method of speech signal parameter modeling and apply it to the modeling of the ECU excitation signal. Om Prakash Yadav et al. [34] proposed a piecewise ECG signal compression model using Chebyshev polynomials for the approximate simulation of each line segment. A. A. Vorontsov et al. modeled the auditory route output signals of magnetostrictive linear or angular displacement sensors mathematically [35]. In view of the above mentioned shortcomings, we propose a segmentation model of the ECU excitation signal based on characteristic parameters (ESCP-SM).

## 3. Characteristic Parameter Analysis

### 3.1. CKP Signal Parameters

The CKP signal is a kind of waveform signal that is used to determine the engine’s speed and crank angle. The CKP signal types $K_{type}$ can be classified as magnetoelectric or Hall according to the type of position sensors. The CKP signal can be divided into two parts: Multitooth signal and missing teeth signal. The total number of teeth $N_{tot}$ is equal to the number of missing teeth $N_{mis}$ plus the number of multiple teeth $N_{mul}$. There are two kinds of missing gear signals used to identify the number of crankshaft revolutions $N_{rev}$. In the first case, $N_{mis}$ can be set to 0. The missing teeth part is simulated by two sinusoidal waveforms with two frequencies. In the second case, the missing tooth waveform is simulated by splicing part of the sine wave, arc, and horizontal line, which are denoted as $s_{1}$, $s_{2}$, and $s_{3}$, respectively. The $N_{mul}$ sinusoidal waves of frequency *f* can simulate the remaining multitooth waveform. For the Hall CKP signal, the missing part can be simulated by the square wave function with a duty ratio of 100%, while the square wave function simulates the other teeth with a duty ratio of 50%. The characteristic parameters of the CKP signal are shown in Table 1. The position of the parameters in the practical signal is shown in Figure 2.

### 3.2. CMP Signal Parameters

The CMP signal is a kind of waveform signal composed of a few pulses synchronized with the CKP signal. The CMP signal types $M_{type}$ can be classified as magnetoelectric or Hall according to the type of position sensors. The spacing of the pulse is consistent except for the marked pulse that is inserted. Determining the position of marked pulse generally requires two parameters: The number of pulses in the preceding interval $N_{int}$ of the marked pulse, and the interval distance $D_{int}$ between the marked pulse and the following pulse. The number of pulse waveforms in the CMP signal is represented by the parameter $N_{pul}$, and the period ratio is μ. For the magnetoelectric CMP signal, the pulse part adopts the ($N_{pul}$− 1) sinusoidal waveform for simulation. The interval part adopts a horizontal line for simulation, and the marked pulse adopts a single sinusoidal waveform for simulation. The position of the marking pulse can be determined by the parameters $N_{int}$ and $D_{int}$. For Hall CMP signals, the teeth are symmetrically spliced by two square waves with a duty ratio of 50%, and square waves simulate the interval part with a duty ratio of 0%. The starting horizontal line offset $L_{offset}$ needs to be set so that the CMP signal is aligned with the CKP signal. The characteristic parameters of the CMP signal are shown in Table 2. The position of the parameters in the practical signal is shown in Figure 2.

## 4. Segmentation Model of ECU Signal

### 4.1. Signal Type and Frequency Parameter Extraction

#### 4.1.1. Determination of Signal Type

To mitigate the effect of individual noise spots on the overall amplitude of the waveform, the data must be normalized. We chose min-max normalization to change the data to a decimal between (0, 1), as shown in Equation (Equation 1):(1)$$x_{norm}=\frac{x-x_{min}}{x_{max}-x_{min}}$$
where $x_{norm}$ represents the normalized data. $x_{min}$ represents the minimum value of the data. $x_{max}$ represents the maximum value of the data.

The types of signals include magnetoelectric and Hall types. The distinction between the two is the change in the difference between adjacent points. Hall signals vary greatly at high and low levels, while magnetoelectric signals vary significantly less. Figure 3 shows the continuous acquisition of different types of CKP and CMP signals.

In Figure 4, the red and blue curves represent the difference values of magnetoelectric signals. By contrast, the green and purple curves represent the different values of Hall signals. The difference in the wave peak of the magnetoelectric signal is below 0.5, and the difference in the wave of the Hall signal is greater than 0.5. In order to reduce the error of noise interference, it is necessary to average the difference wave peaks that meet a certain threshold. The average peak value is calculated as follows:(2)$$\xi =\frac{\Sigma Peak(x_{norm})}{N_{peak}},x_{norm}>\tau$$
where ξ represents the average of the peak. τ represents the threshold value of peak search, and peak detection is performed only when the point is greater than this threshold value. $Peak(x_{norm})$ represents the peak detection function, and the maximum value of the local waveform is obtained. $x_{norm}$ represents the normalized data. $N_{peak}$ represents the number of peaks.

The value of τ is set to 0.2 for wave peak detection. If $N_{peak}$ is equal to 0, ξ is assigned to 0. When ξ is greater than 0.5, it is judged to be a Hall signal. Otherwise, it is a magnetoelectric signal.

#### 4.1.2. Signal Frequency Extraction

Frequency is a key global parameter of the signal, and it can determine the vehicle’s speed information. Discrete Fourier transform (DFT) is used to extract the frequency domain features of the signal. The formula for the DFT is shown below:(3)$$X\left(k\right)=\sum_{n=0}^{N-1}x\left(n\right)e^{-j\frac{2\pi kn}{N}}$$
where *N* is the Fourier transform points, and *k* is the kth spectrum of the Fourier transform. X(k) represents the data after DFT transformation, and x(n) represents the actual signal sampled.

The larger the amplitude of *k* in the spectrum, the more frequencies corresponding to this *k* value are contained in the signal. However, the *k* value is not the actual signal’s frequency, and the relationship between them is shown in Equation (Equation 4).
(4)$$f=k\frac{Fs}{N_{fft}}$$
where *k* is the kth spectrum of the Fourier transform. *f* is the frequency of the actual signal. $N_{fft}$ is the Fourier transform points. $F_{s}$ represents the sampling rate, which is the number of points of signal collected per second.

As shown in Figure 5, the spectrum of magnetoelectric CKP signals and Hall CKP signals are analyzed using discrete Fourier transform (DFT). The oscilloscope is used to collect 1000 points within 0.1 s, so it can be concluded that the value of Fs is 10,000, and the point $N_{fft}$ of the Fourier transform is 1000 HZ. In the spectrum, the amplitude is greatest when *k* is 100. Therefore, the frequency of the actual signal can be calculated as 1000 HZ.

### 4.2. Modeling Process

The CKP signal modeling process is detailed in Figure 6. The CKP signal is simulated according to the characteristic parameters. The frequency of the signal is set according to the fast Fourier transform. Firstly, load the CKP parameters file and initialize the characteristic parameters. Secondly, set and initialize global variables, including signal period *T* and duty cycle $\lambda_{duty}$. Thirdly, the type of CKP signal is determined. If the $K_{type}$’ value is 0, it is a magnetoelectric signal; otherwise, it is a Hall signal. Secondly, the multitooth waveform $W_{multiteeth}$ and the missing tooth waveform $W_{missing}$ are generated. The $W_{multiteeth}$ are generated by the sinusoidal waveform of $N_{mul}$ cycles. According to the number of missing teeth $N_{mis}$, different types of missing tooth waves are generated, including missing tooth wave and no missing tooth wave $W_{no\_missing}$. Finally, the CKP signal waveform $W_{CKP}$ is offset upward.

The CMP signal modeling process is detailed in Figure 7. The CMP signal is simulated according to the characteristic parameters. Firstly, load the CMP parameters file and initialize the characteristic parameters. Secondly, set and initialize global variables, including signal period *T* and duty cycle $\lambda_{duty}$. Thirdly, the length (*L*) of the horizontal line waveform $W_{line}$ between pulses based on μ, $N_{pul}$, and other parameters is calculated. Fourthly, different types of pulse waveforms $W_{pul}$ are generated according to the value of parameter $M_{type}$. When the $M_{type}$’ value is 0, the pulse waveforms are generated by the sinusoidal waveform of $N_{pul}$ cycles. When the $M_{type}$’ value is not 0, the pulse waveforms are generated by two opposite symmetrical square wave shapes. Fifthly, the position of the marked pulse waveform $W_{mark\_pulse}$ is determined. Finally, the generated CMP signal waveform $W_{CMP}$ is offset upward.

### 4.3. Magnetoelectric CKP Signal Model

The magnetoelectric CKP signal can be divided into a missing teeth CKP waveform and no-missing-teeth CKP waveform according to the number of missing teeth. The magnetoelectric CKP signal can also be generally divided into two sections: The multitooth waveform and the missing tooth waveform. Usually, the waveform generated by the function is continuous, but the data sent to the data acquisition card are discrete. It is necessary to multiply the independent variable of the function by reciprocating the sampling rate to generate discrete waveforms at equal intervals. The ratio of the sampling rate $F_{s}$ to the frequency *f* is the number of points in a single period.

#### 4.3.1. No-Missing-Teeth Two-Stage Modeling

The no-missing-teeth CKP signal comprises two sections: The multitooth signal and the marked tooth signal.

Stage 1: Multitooth Modeling. A sinusoidal waveform of $N_{mul}$ cycles generates the multitooth waveform, as shown in Equation (Equation 5):(5)$$y_{mk\_zmp}\left(x\right)={\left(-1\right)}^{i_{K}}Asin\left(2\pi fx\right)+V_{offset\_K},x\in \left[0,\frac{N_{mul}}{f}\right)$$
where $y_{mk\_zmp}\left(x\right)$ represents the multitooth waveform of the no-missing-teeth signal. The independent variable *x* represents a sequence of sampling points. $\frac{N_{mul}}{f}$ represents a sine wave with $N_{mul}$ periods.

Stage 2: Marked tooth Modeling. The two sinusoidal waveforms, the period of which is half of the sinusoidal waveform of the multitooth signal, generate the marked tooth signal, as shown in Equation (Equation 6):(6)$$y_{mar}\left(x\right)={\left(-1\right)}^{i_{K}}Asin\left(4\pi f\left(x-\frac{N_{mul}}{f}\right)\right)+V_{offset\_K},x\in \left[\frac{N_{mul}}{f},\frac{N_{mul}+1}{f}\right)$$
where $y_{mar}\left(x\right)$ represents the marked tooth waveform of the no-missing-teeth signal. $\left(x-\frac{N_{mul}}{f}\right)$ represents that the marked tooth waveform needs to be shifted to the right for the $N_{mul}$ period to be spliced behind the multitooth waveform. The mark tooth signal period is exactly a sinusoidal period in the multi-teeth signal, so the domain needs to add periodic sampling points.

The complete no-missing-teeth magnetoelectric CKP signal is shown in Figure 8.

#### 4.3.2. Missing Teeth Six-Stage Modeling

The missing teeth CKP signal comprises six sections: The multitooth signal, the front sine part, the front circular arc part, the middle horizontal line part, the behind circular arc part, and the behind sine part. The last five components constitute the missing tooth waveform.

Stage 1: Multitooth modeling. This stage is the same as the first stage of the no-missing-teeth CKP modeling.
(7)$$y_{mk\_mp}\left(x\right)={\left(-1\right)}^{i_{K}}Asin\left(2\pi fx\right)+V_{offset\_K},x\in \left[0,\frac{N_{mul}}{f}\right)$$
where $y_{mk\_mp}\left(x\right)$ represents the multitooth waveform of the missing teeth signal. The independent variable *x* represents a sequence of sampling points. $\frac{N_{mul}}{f}$ represents a sine wave with $N_{mul}$ periods.

Stage 2: Front missing tooth sine modeling. For the front sinusoidal part of the missing teeth CKP signal, the sinusoidal waveform of $s_{1}$ times the period is generated according to Equation (Equation 8):(8)$$y_{f\_sin}\left(x\right)={\left(-1\right)}^{i_{K}}Asin\left(2\pi f\left(x-\frac{N_{mul}}{f}\right)\right)+V_{offset\_K},x\in \left[\frac{N_{mul}}{f},\frac{N_{mul}+s_{1}}{f}\right)$$
where $y_{f\_sin}\left(x\right)$ represents the front sinusoidal part of the missing teeth signal. The front sinusoidal part period is $s_{1}$ times period, so the domain needs to add $s_{1}$ times the periodic sampling points.

Stage 3: Front missing tooth arc modeling. The sinusoidal part and arc part of the missing tooth waveform are $s_{1}$ times and $s_{2}$ times the sinusoidal function period, and then the remaining horizontal line part is $s_{3}$, given as follows:(9)$$s_{3}=\frac{N_{mis}+1}{2}-s_{1}-s_{2}.$$

The sinusoidal part and the horizontal line part of the missing teeth waveform are connected smoothly by the circular part. Suppose the radius of the circle is *r*, then the center coordinate of the circle is $(r+\frac{s_{1}}{f},$ b). The last value of the sine part is *b*.
(10)$$b=Asin(2\pi s_{1})$$
where the independent variable of the last point of the sine part is $\frac{s_{1}}{f}$. After substituting b=Asin(2πfx), *f* is cancelled out:(11)$$r=\frac{s_{2}}{2f}+\frac{b^{2}f}{2s_{2}}.$$

For the front arc part of the missing teeth CKP signal, the arc waveform of $s_{2}$ times the period is generated according to Equation (Equation 12):(12)$$\begin{array}{c} y_{f\_arc}\left(x\right)={\left(-1\right)}^{i_{K}}\left(\sqrt{2r\left(x-\frac{N_{mul}+s_{1}}{f}\right)-{\left(x-\frac{N_{mul}+s_{1}}{f}\right)}^{2}}+b\right)+V_{offset\_K},\\ x\in [\frac{N_{mul}+s_{1}}{f},\frac{N_{mul}+s_{1}+s_{2}}{f}) \end{array}$$
where $y_{f\_arc}\left(x\right)$ represents the front arc part of the missing teeth signal.

The first half of the missing teeth waveform is generated by Equations (Equation 8) and (Equation 12). In Figure 9, circle 1 in blue represents the sinusoidal portion, and circle 2 in red represents the arc portion.

Stage 4: Middle missing tooth line modeling. For the middle horizontal line of the missing teeth CKP signal, the horizontal line with $s_{3}$ times the period is generated according to Equation (Equation 13):(13)$$\begin{array}{c} y_{m\_line}\left(x\right)=0+V_{offset\_K},x\in \left[\frac{N_{mul}+s_{1}+s_{2}}{f},\frac{N_{tot}-s_{1}-s_{2}}{f}\right) \end{array}$$
where $y_{m\_line}\left(x\right)$ represents the middle horizontal line of the missing teeth signal.

Stage 5: Behind missing tooth arc modeling. For the behind arc part of the missing teeth CKP signal, the center coordinate of the circle is $(\frac{s_{1}}{f}-r,-$ b). The circular arc waveform with $s_{2}$ times the period is generated according to Equation (Equation 14):(14)$$\begin{array}{c} y_{b\_arc}\left(x\right)={\left(-1\right)}^{i_{K}}\left(\sqrt{r^{2}-{\left(x-\frac{N_{tot}-s_{1}}{f}+r\right)}^{2}}-b\right)+V_{offset\_K},\\ x\in [\frac{N_{tot}-s_{1}-s_{2}}{f},\frac{N_{tot}-s_{1}}{f}) \end{array}$$
where $y_{b\_arc}\left(x\right)$ represents the behind arc part of the missing teeth signal.

Stage 6: Behind missing tooth sine modeling. For the behind sinusoidal part of the missing teeth CKP signal, the sinusoidal waveform of $s_{1}$ times the period is generated according to Equation (Equation 15).
(15)$$\begin{array}{c} y_{b\_sin}\left(x\right)={\left(-1\right)}^{i_{K}}Asin\left(2\pi f\left(x-\frac{N_{mul}+N_{mis}}{f}\right)\right)+V_{offset\_K},x\in \left[\frac{N_{tot}-s_{1}}{f},\frac{N_{tot}}{f}\right) \end{array}$$
where $y_{b\_sin}\left(x\right)$ represents the behind sinusoidal part of the missing teeth signal.

The posterior part of the missing tooth waveform is generated by Equations (Equation 14) and (Equation 15). In Figure 10, circle 1 in red represents the arc portion, and circle 2 in blue represents the sinusoidal portion.

The complete missing teeth magnetoelectric CKP signal is shown in Figure 11. The whole part includes multitooth waveform and missing tooth waveform.

### 4.4. Hall CKP Signal Model

As shown in Figure 12, the Hall CKP signal can be generally divided into two sections: The multitooth waveform and the missing tooth waveform.

The function square(p) is defined as:(16)$$square\left(p\right)=\left\{\begin{array}{ccc} 1 & \; & 0\leq p\;\%\;2\pi <2\pi \lambda_{duty}\\ -1 & \; & 2\pi \lambda_{duty}\leq p\;\%\;2\pi <2\pi \end{array}\right.$$
where % is the modulo operation and $\lambda_{duty}$ is the duty ratio.

Stage 1: Multitooth modeling. In the first stage of the Hall CKP signal, the square multitooth waveform part with $N_{mul}$ cycles is generated according to Equation (Equation 17).
(17)$$y_{hk\_mmp}\left(x\right)={\left(-1\right)}^{i_{K}}Asquare\left(2\pi fx\right)+V_{offset\_K},\lambda_{duty}=0.5,x\in \left[0,\frac{N_{mul}}{f}\right)$$
where $y_{hk\_mmp}\left(x\right)$ represents the square multitooth waveform.

Stage 2: Missing teeth modeling. In the second stage of the Hall CKP signal, the square missing tooth waveform part with ($N_{mis}$ +1) cycles is generated according to Equation (Equation 18):(18)$$y_{hk\_mmp}\left(x\right)={\left(-1\right)}^{i_{K}}Asquare\left(2\pi fx\right)+V_{offset\_K},\lambda_{duty}=1,x\in \left[\frac{N_{mul}}{f},\frac{N_{tot}}{f}\right).$$

### 4.5. Magnetoelectric CMP Signal Model

The magnetoelectric CMP signal consists of three parts: A sinusoidal pulse waveform, horizontal line waveform, and sinusoidal marked pulse waveform.

Stage 1: Sine pulse modeling. The period of the sinusoidal pulse waveform in the CMP signal is μT; then, the sinusoidal pulse signal in the CMP waveform is generated according to Equation (Equation 19).
(19)$$\begin{array}{c} y_{mm\_sp}\left(x\right)={\left(-1\right)}^{i_{K}}Asin\left(\frac{2\pi f}{\mu }\left(x-\left(\frac{kL+(k-1)\mu }{f}\%\frac{\mu }{f}\right)\right)\right)+V_{offset\_K},\\ x\in \left[\frac{kL+(k-1)\mu }{f},\frac{k(L+\mu )}{f}\right),k=1,2,\ldots ,N_{pul}-1 \end{array}$$
where $y_{mm\_sp}\left(x\right)$ represents the sinusoidal pulse waveform. μ is the periodic multiples of the pulse signal. The modulo operation (%) ensures that the sinusoidal impulse is still a complete sine after being shifted by a certain unit. The meanings of other variables are shown in Table 2.

Stage 2: Horizontal line modeling. The horizontal line waveform is generated according to Equation (Equation 20).
(20)$$y_{mm\_hl}\left(x\right)=0+V_{offset\_K},x\in \left[\frac{\left(k-1)(L+\mu \right)}{f},\frac{kL+(k-1)\mu }{f}\right),k=1,2,\ldots ,N_{pul}-1$$
where $y_{mm\_hl}\left(x\right)$ represents the horizontal line waveform.

The length *L* of the horizontal line is calculated according to Equation (Equation 21):(21)$$L=\frac{N_{tot}N_{rev}-\mu \left(N_{pul}-1\right)}{(N_{pul}-1)f}$$
where $N_{tot}$ is the number of teeth of the gear. $N_{rev}$ is the number of revolutions of the crankshaft.

Stage 3: Sinusoidal marked pulse modeling. One of the sinusoidal waveforms is the marked pulse waveform, and the interval length between the other sinusoidal waveforms is the same. The sinusoidal marked pulse waveform is generated according to Equation (Equation 22).
(22)$$\begin{array}{c} y_{mm\_mp}\left(x\right)={\left(-1\right)}^{i_{M}}Asin\left(2\pi f\left(x-\left(\frac{\left(N_{int}+1\right)L+\left(N_{int}-1\right)\mu -D_{int}}{f}\right)\%\frac{u}{f}\right)\right)+V_{offset\_M},\\ x\in [\frac{\left(N_{int}+1\right)L+\left(N_{int}-1\right)\mu -D_{int}}{f},\frac{\left(N_{int}+1\right)L+N_{int}\mu -D_{int}}{f}) \end{array}$$
where $y_{mm\_mp}\left(x\right)$ represents the sinusoidal marked pulse waveform.

The complete magnetoelectric CMP signal is shown in Figure 13.

### 4.6. Hall CMP Signal Model

The Hall CMP signal consists of three parts: Rectangular pulse waveform, horizontal line waveform, and square marked pulse waveform.

Stage 1: Rectangular pulse Modeling. The front part of the rectangular pulse consists of a square wave with a duty ratio $\lambda_{duty}=50\%$, the middle part consists of a square wave with a duty ratio $\lambda_{duty}=100\%$, and the rear part consists of the front part turning 180° horizontally.

When μ>1, the rectangular pulse in the CMP signal is generated according to Equation (Equation 23):(23)$$y_{hm\_hp}\left(x\right)\left\{\begin{array}{ccc} -Asquare\left(2\pi f\left(x-\left(\frac{kL+(k-1)\mu }{f}\%\frac{1}{f}\right)\right)\right)+V_{offset\_M}, & \lambda_{duty}=0.5, & x\in [\frac{kL+(k-1)\mu }{f},\frac{kL+(k-1)\mu +1}{f})\\ Asquare\left(2\pi f\left(x-\left(\frac{kL+(k-1)\mu +1}{f}\%\frac{1}{f}\right)\right)\right)+V_{offset\_M}, & \lambda_{duty}=1, & x\in [\frac{kL+(k-1)\mu +1}{f},\frac{k(L+\mu )-1}{f})\\ Asquare(2\pi f\left(x-\left(\frac{k(L+\mu )-1}{f}\%\frac{1}{f}\right)\right)+V_{offset\_M}, & \lambda_{duty}=0.5, & x\in [\frac{k(L+\mu )-1}{f},\frac{k(L+\mu )}{f}). \end{array}\right.$$

The rectangular pulse is shown in Figure 14.

Stage 2: Horizontal line Modeling. The voltage amplitude of the horizontal line is 0. The length of the horizontal line and the marked pulse signal’s position parameters are the same as that of the magnetoelectric CMP signal.

The horizontal line waveform is generated according to Equation (Equation 24).
(24)$$\begin{array}{c} y_{hm\_hl}\left(x\right)={\left(-1\right)}^{i_{M}}Asquare\left(2\pi fx\right)+V_{offset\_M},\lambda_{duty}=0,\\ x\in \left[\frac{\left(k-1)(L+\mu \right)}{f},\frac{kL+(k-1)\mu }{f}\right),k=1,2,\ldots ,N_{pul}-1 \end{array}$$
where, $y_{hm\_hl}\left(x\right)$ represents the horizontal line waveform.

Stage 3: Marked pulse modeling. The rectangular marked pulse waveform is generated according to Equation (Equation 25):(25)$$y_{hm\_mp}\left(x\right)\left\{\begin{array}{cc} -Asquare\left(2\pi f\left(x-\left(\frac{\left(N_{int}+1\right)L+\left(N_{int}-1\right)\mu -D_{int}}{f}\%\frac{1}{f}\right)\right)\right)+V_{offset\_M}, & \lambda_{duty}=0.5\\ Asquare\left(2\pi f\left(x-\left(\frac{\left(N_{int}+1\right)L+\left(N_{int}-1\right)\mu -D_{int}+1}{f}\%\frac{1}{f}\right)\right)\right)+V_{offset\_M}, & \lambda_{duty}=1\\ Asquare(2\pi f\left(x-\left(\frac{\left(N_{int}+1\right)L+N_{int}-1\mu -D_{int}-1}{f}\%\frac{1}{f}\right)\right)+V_{offset\_M}, & \lambda_{duty}=0.5. \end{array}\right.$$

The complete Hall CMP signal is shown in Figure 15.

## 5. Software Development and Experiment

### 5.1. Signal Generator Software Development

According to the above modeling method, we designed a signal generator software. This software included four functional modules: The loading parameter file module, output channel selection module, rotating speed frequency modulation module, and simulation signal display module. The software can generate a variety of CKP and CMP signals so that the construction vehicle’s ECU can be detected offline. To improve the software’s extensibility, we designed the signal parameter editing page to meet the actual test requirements. The main interface of the signal generator software is shown in Figure 16. The CKP and CMP signals for different vehicle models are shown in Figure 17.

### 5.2. Simulation Experiment

The ECU excitation signal segment model with the above characteristic parameter configuration was realized by programming the previously mentioned computer software. The magnetoelectric CKP signal was taken as an example to generate the signal of the corresponding parameter configuration. In terms of the hardware circuit simulation, the HBs-EED method was selected to compare with the real vehicle signal. In relation to the virtual instrument simulation, the MSS-AS method was chosen to compare with the actual collected signal.

In Figure 18, the blue curve represents the actually collected CKP signal. The red curve represents the ESCP-SM simulation CKP signal. The actually collected CKP signal vibrated more, which was caused by the interference of other signals during the acquisition process.

In Figure 19, the blue curve represents the actually collected CKP signal. The red curve represents the HBS-EED simulation CKP signal. In the HBS-EED simulation, the CKP signal vibration is large, and the signal amplitude fluctuates wildly.

In Figure 20, the blue curve represents the actually collected CKP signal. The red curve represents the MSS-AS simulation CKP signal. The signal simulated by the MMS-AS method is simulated by the arcsine function at the missing tooth waveform, and the waveform is relatively steep.

In Figure 21, the simulated signal of our proposed ESCP-SM method is most similar to the actually collected signal. The signal voltage simulated by the HBS-EED method is not stable. The signal simulated by the MMS-AS method is relatively steep in the missing tooth waveform, and can not be very close to the actual signal.

To verify the parameter configuration model’s accuracy, the Pearson similarity formula [36] was used to obtain the correlation coefficient, and the similarity between the simulation CKP signal and the actually collected CKP signal was analyzed. The similarity coefficient $\rho_{X,Y}$ can be calculated by Equation (Equation 26):(26)$$\rho_{X,Y}=\frac{\sum \left(X-\bar{X}\right)\left(Y-\bar{Y}\right)}{\sqrt{\sum {\left(X-\bar{X}\right)}^{2}{\left(Y-\bar{Y}\right)}^{2}}}$$
where *X* and *Y* represent two groups of data, $\bar{X}$ represents the average value of the data of group *X*, and $\bar{Y}$ and represents the average value of the data of group *Y*.

According to the Pearson correlation coefficient’s value, we can estimate the degree of similarity between the two. By comparing the similarity between the ESCP-SM simulation CKP signals and the actual CKP signals, the final result was 0.74, which indicated a strong correlation. By comparing the similarity between the HBS-EED-simulated CKP signal and the actual CKP signal, the final result was 0.48. In the HBS-EED simulation, the CKP signal vibration is large due to the voltage instability of circuit components. The amplitude stabilizing circuit leads to difficulty in controlling the positive feedback, resulting in the amplitude of the sine wave fluctuating. Compared to the HBS-EED simulation method, our ESCP-SM simulation based on the modeling technique improved the accuracy by 26%. By comparing the similarity between the MSS-AS simulated CKP signal and actual CKP signal, the final result is 0.68. The MSS-AS method uses the arcsine function to generate the missing tooth waveform signal, which is relatively steep. Compared with the proposed ESCP-SM method, the signal accuracy is improved by 6%.

Since the environment of the engine test bench is difficult to build, we can only go to the real vehicle to collect signal data. The real vehicle data usually collected is easily affected by noise and uneven engine speed and other factors, resulting in a certain deviation of our simulated signal when compared. In fact, our simulated signal is much closer to the ideal signal, with an accuracy of more than 74%.

To verify that the simulation CKP signal’s effective information was not lost, the actual CKP signals and simulation CKP signals were analyzed with a fast Fourier spectrum [37]. Observation of spectrum analysis shows that the software simulation signal’s effective information is not lost, while the hardware simulation signal adds a redundant noise signal. The spectrum analysis diagram is shown in Figure 22.

### 5.3. Physical Experiment

The experimental environment was set up. The ESCP-SM was realized based on virtual instrument technology and provided an excitation signal for the ECU through the data acquisition (DAQ) card. The voltage-stabilized source supplied the ECU with power and the ignition switch signal. The experimental environment is shown in Figure 23.

Collecting the injection pulse signal output by the ECU with an oscilloscope can detect whether the ECU can recognize the simulated CKP and CMP signals. The ECU can recognize the signal normally and issue the injection instruction after the actual test. The injection waveform output of the ECU is shown in Figure 24. The PWM wave in the figure below is the injection duration.

## 6. Conclusions

This study analyzed the characteristic parameters of the ECU excitation signal, and a segmentation model of the ECU excitation signal based on characteristic parameters is proposed in this paper. The ESCP-SM is more dynamic and scalable, which leads to an improved complex excitation signal simulation. The experimental results show that the proposed ESCP-SM method is much better than the hardware simulation HBS-EED model in the similarity detection of signals collected from actual vehicle sensors. By the parameterized configuration, the diversity of signals is significantly increased.

The model can not only be applied to diesel vehicles but can also be extended to gasoline vehicles by adjusting the parameters. For part of the automotive camshaft signal, additional parameters of the proportional relationship between different protruding pulses are needed. For other excitation signals of ECUs, such as oxygen sensor signals, we need to add some additional uniform white noise signals to simulate actual sensor signals while adjusting characteristic parameters. Researchers in related fields can draw on the ideas of this model to develop corresponding simulation signals. First, the intrinsic characteristic parameters of the signals are extracted and abstracted, and then appropriate mathematical functions are used to piecemeal fit local signals. The accuracy of the simulation signal can be significantly improved by using this model.

When applied to the actual ECU test of engineering vehicles in the later stage, we need to burn the program into the development board and separately design the corresponding data acquisition card to reduce the test cost. The released open source code can help researchers in related fields to provide some ideas for reference when designing multi-type and complex parameter simulation signals. In future studies, we can further use more accurate characteristic parameters to simulate irregular signals. 

## Figures and Tables

**Figure 1 sensors-21-04165-f001:**
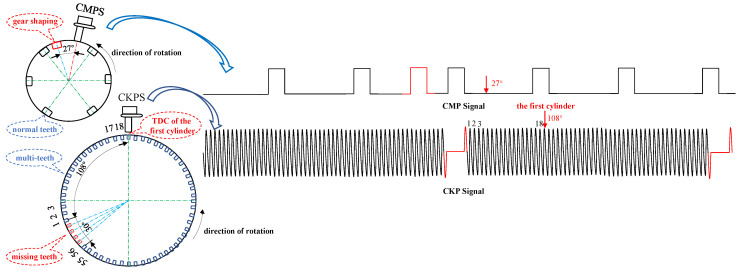
The crankshaft and camshaft joint judgment cylinder diagram. Through the CKP and CMP signals, the ECU determines the top dead center of a cylinder and then makes the cylinder sequence determination. CMPS: Camshaft position sensor. CKPS: Crankshaft position sensor. TDC: Top dead center.

**Figure 2 sensors-21-04165-f002:**
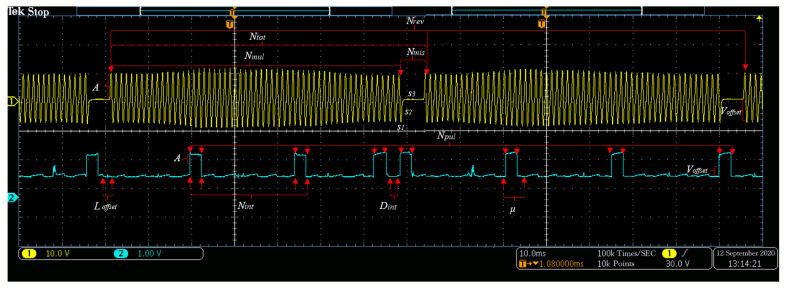
The CKP and CMP signal parameters diagram. The yellow curve represents the actual collected CKP signal. The blue curve represents the actual collected CKP signal.

**Figure 3 sensors-21-04165-f003:**
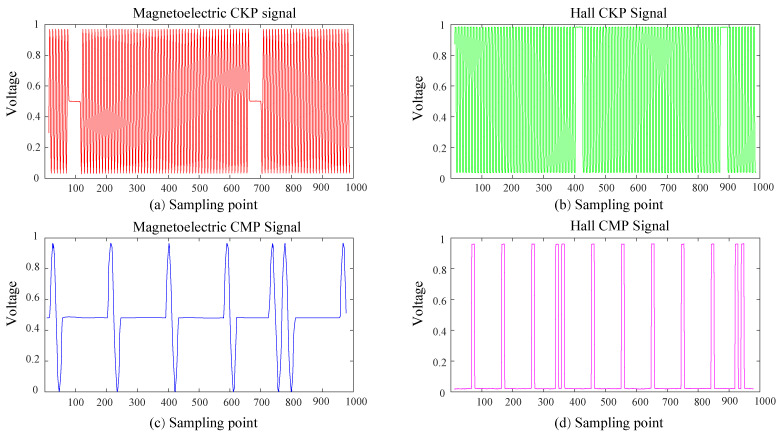
The different types of CKP and CMP signal diagram. The red and blue curves represent magnetoelectric signals that are shaped like sinusoidal waveforms. The green and pink curves represent magnetoelectric signals that are shaped like rectangular waveforms.

**Figure 4 sensors-21-04165-f004:**
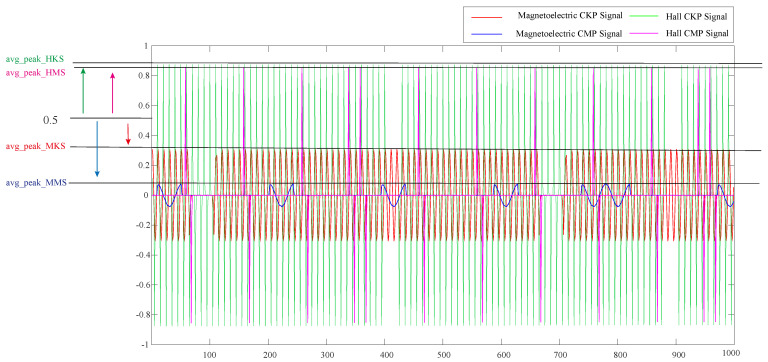
The adjacent point difference diagram. avg_peak_HKS, avg_peak_HMS, avg_peak_MKS, and avg_peak_MKS represent the mean values of signal peaks respectively.

**Figure 5 sensors-21-04165-f005:**
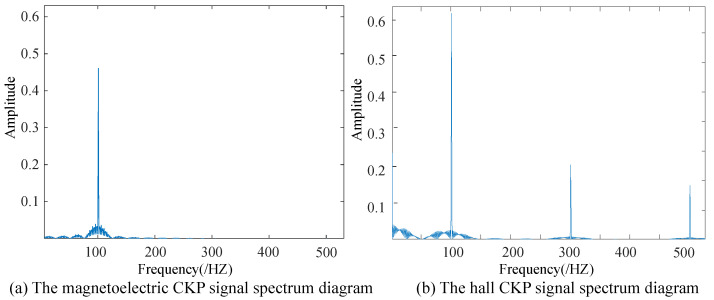
The CKP signal spectrum diagram. Magnetoelectric CKP signals have a maximum spectrum value of *k* at 100. The Hall CKP signal has a maximum spectrum value of *k* at 100.

**Figure 6 sensors-21-04165-f006:**
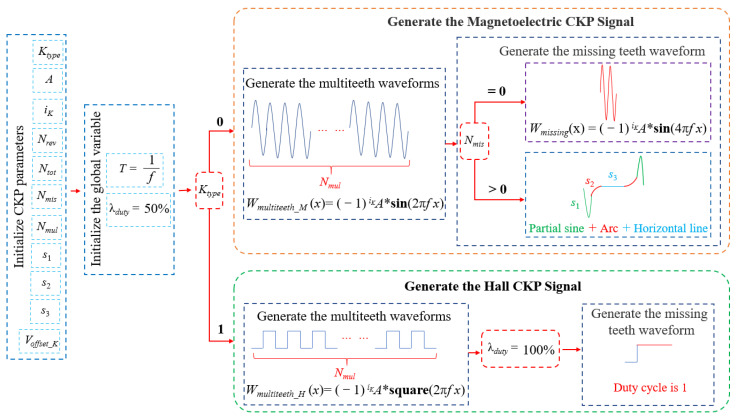
The CKP signal modeling process. *T* is the period of the signal. *f* is the frequency of the signal. $\lambda_{duty}$ represents the duty cycle of the square wave signal.

**Figure 7 sensors-21-04165-f007:**
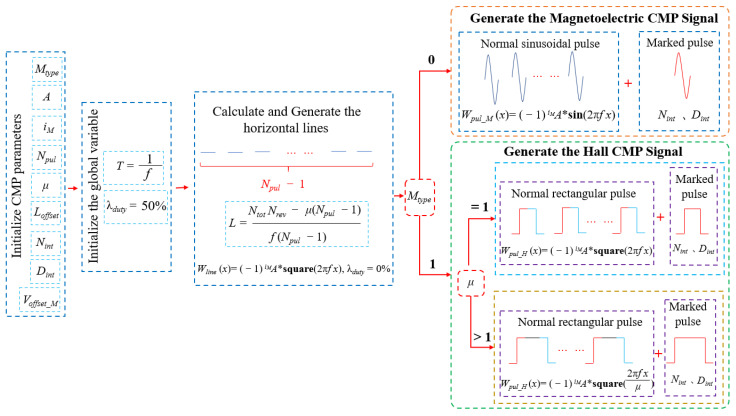
The CMP signal modeling process. *L* represents the length of a single horizontal line signal. μ represents the period multiple of a single pulse signal.

**Figure 8 sensors-21-04165-f008:**
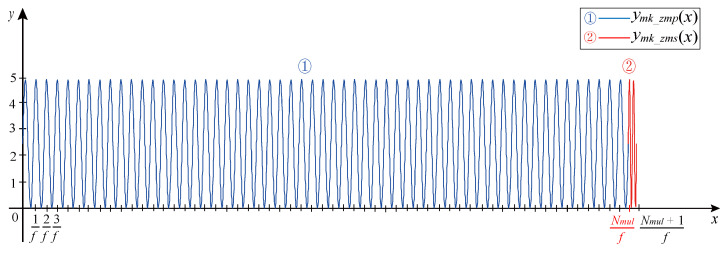
No missing teeth magnetoelectric CKP signal.

**Figure 9 sensors-21-04165-f009:**
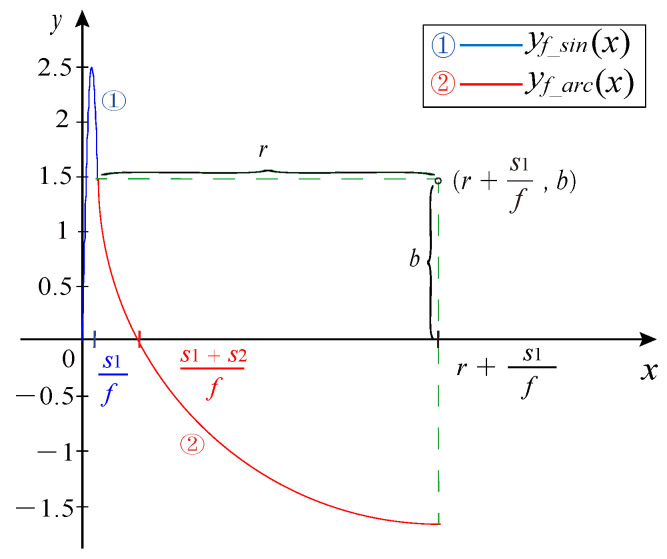
The first half of the missing waveform.

**Figure 10 sensors-21-04165-f010:**
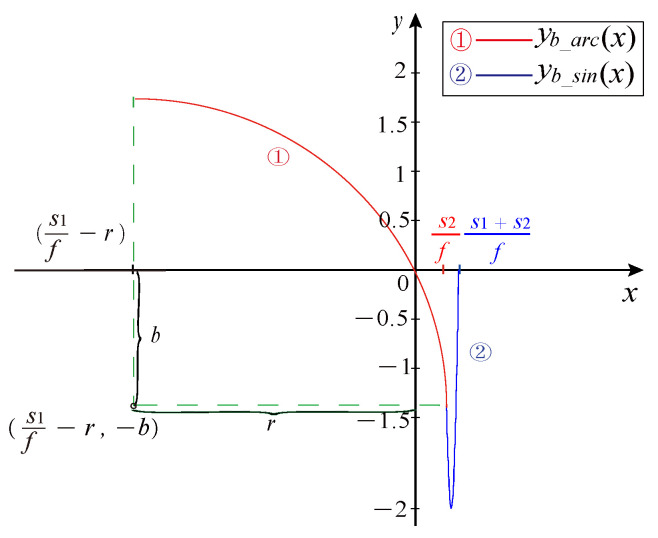
The posterior part of the missing waveform.

**Figure 11 sensors-21-04165-f011:**
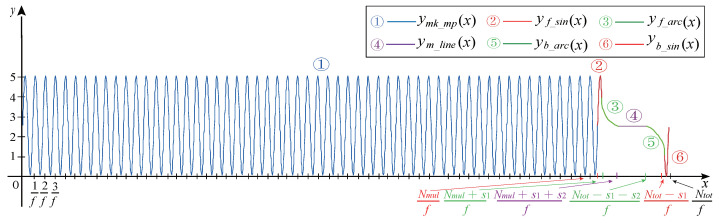
Missing teeth magnetoelectric CKP signal.

**Figure 12 sensors-21-04165-f012:**
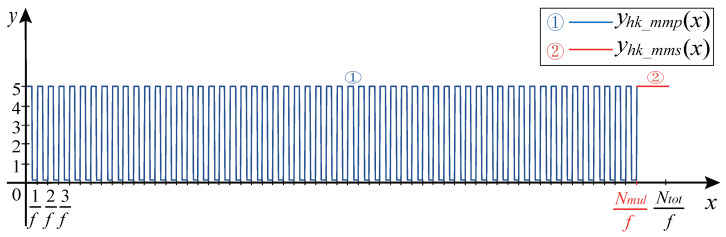
Hall CKP signal.

**Figure 13 sensors-21-04165-f013:**
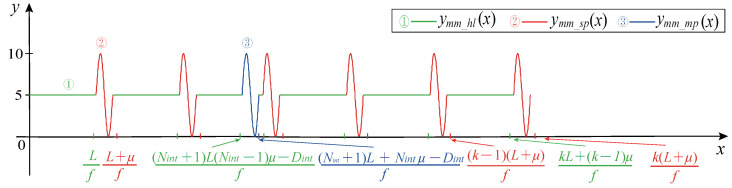
Magnetoelectric CMP signal.

**Figure 14 sensors-21-04165-f014:**
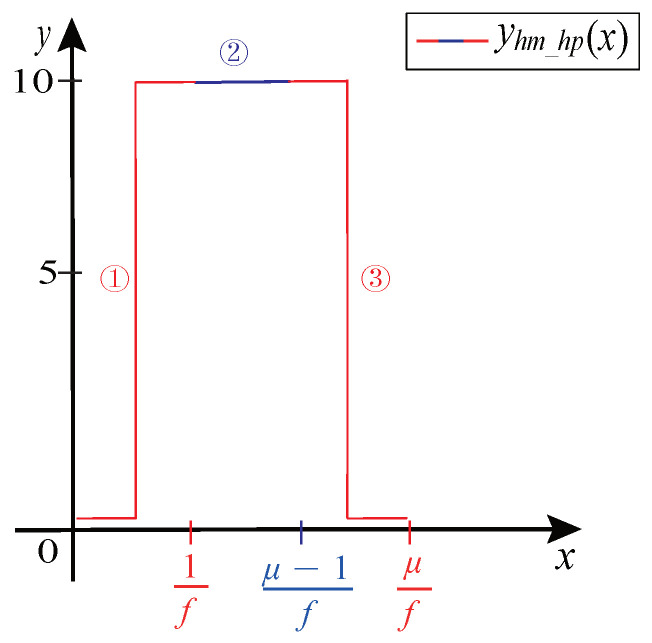
The rectangular pulse.

**Figure 15 sensors-21-04165-f015:**
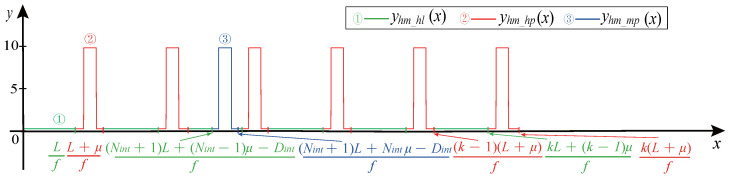
Hall CMP signal.

**Figure 16 sensors-21-04165-f016:**
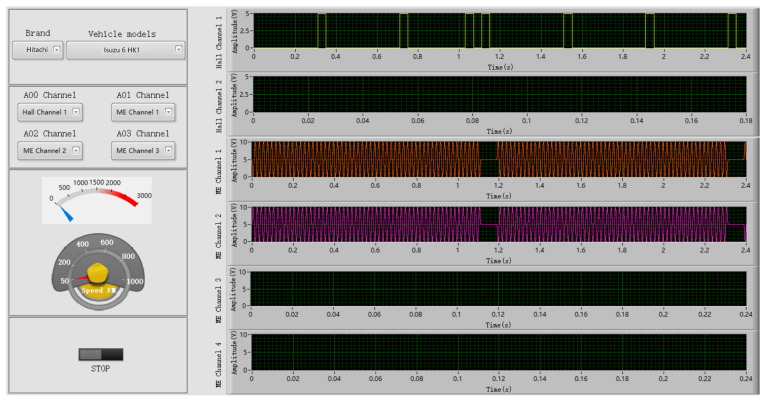
The main interface of signal generator software.

**Figure 17 sensors-21-04165-f017:**
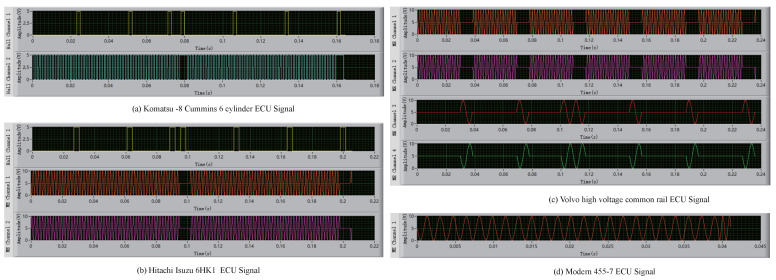
Excitation signals for four types of ECU.

**Figure 18 sensors-21-04165-f018:**
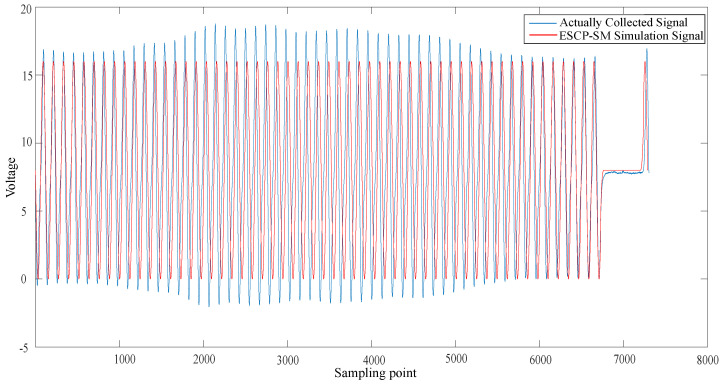
Comparison of the ESCP-SM simulation and actually collected signal.

**Figure 19 sensors-21-04165-f019:**
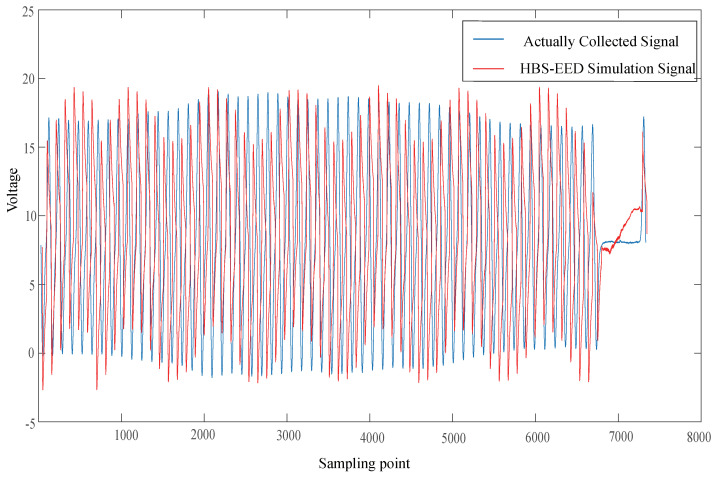
Comparison of the HBS-EED simulation and actually collected signal.

**Figure 20 sensors-21-04165-f020:**
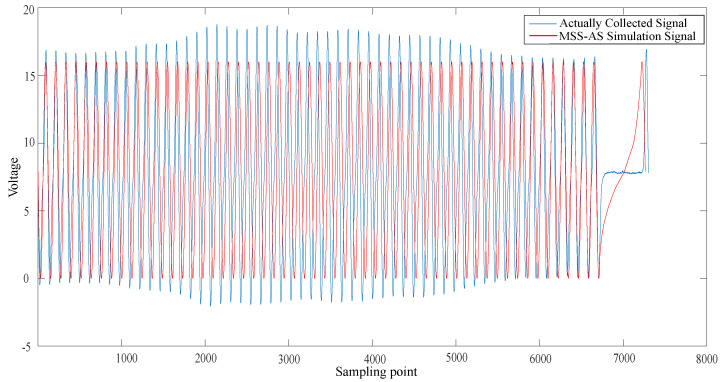
Comparison of the MSS-AS simulation and actually collected signal.

**Figure 21 sensors-21-04165-f021:**
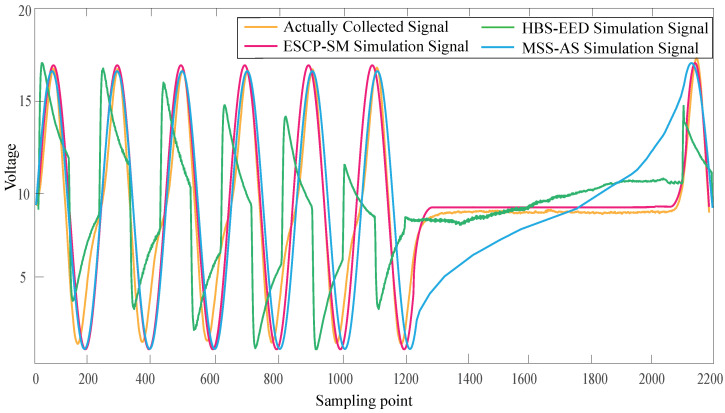
Local magnification of the four signals.

**Figure 22 sensors-21-04165-f022:**
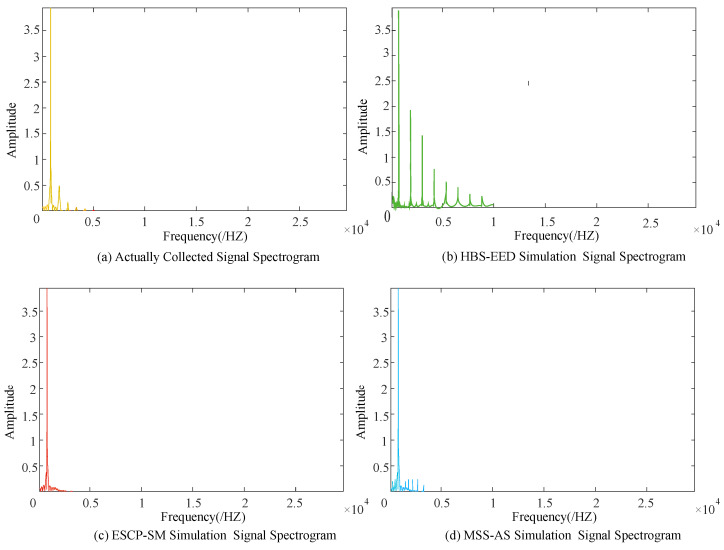
Spectrum analysis diagram.

**Figure 23 sensors-21-04165-f023:**
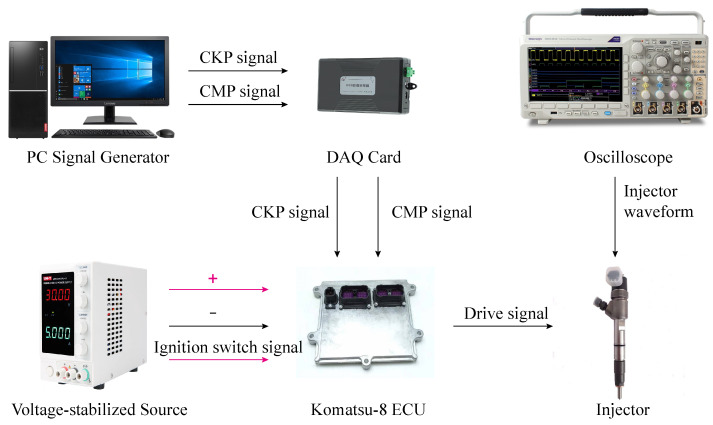
Setting up experimental environment.

**Figure 24 sensors-21-04165-f024:**
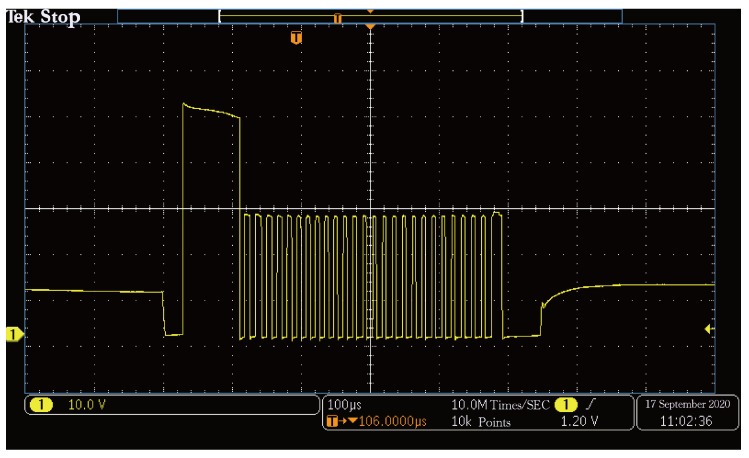
Injection pulse signal diagram.

**Table 1 sensors-21-04165-t001:** CKP signal characteristic parameters.

SN	Parameters	Range	Meaning
1	$K_{type}$	{0, 1}	CKP signal type
2	*A*	(0, 10]	Voltage amplitude
3	$i_{K}$	{0, 1}	Signal inversion
4	$N_{rev}$	$N_{+}$	Number of revolution
5	$N_{tot}$	$N_{+}$	Total number of teeth
6	$N_{mis}$	$N_{+}$	Number of missing teeth
7	$N_{mul}$	$N_{tot}$-$N_{mis}$-1	Number of multiteeth
8	$s_{1}$	(0, 0.5)	Partial sinusoidal multiple
9	$s_{2}$	(0, 0.5)	Partial arc multiple
10	$s_{3}$	$N_{mis}$-$s_{1}$-$s_{2}$	Partial line multiple
11	$V_{offset\_K}$	$N_{+}$	Voltage offset of CKP signal

**Table 2 sensors-21-04165-t002:** CMP signal characteristic parameters.

SN	Parameters	Range	Meaning
1	$M_{type}$	{0, 1}	CMP signal type
2	*A*	(0, 10]	Voltage amplitude
3	$i_{M}$	{0, 1}	Signal inversion
4	$N_{pul}$	$N_{+}$	Number of pulses
5	μ	$N_{+}$	Periodic multiples
6	$L_{offset}$	$N_{+}$	The starting horizontal line offset
7	$N_{int}$	$N_{+}$	Number of pulses in the interval
8	$D_{int}$	$N_{+}$	The interval distance
9	$V_{offset\_M}$	$N_{+}$	Voltage offset of CMP signal

## Data Availability

The datasets generated from the current study are available from the corresponding author on reasonable request.

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
