# Peer review of "A Segmentation Model of ECU Excitation Signal Based on Characteristic Parameters"

_sensors, 2021, doi:10.3390/s21124165_

Round 1
Reviewer 1 Report
Dear Authors,
I have some comments on your article:
- At the end of the introduction section, there is no information on how the article is organized.
- Process 2 Modeling Process for CMP Signal and Process 1 Modeling Process for CKP Signal - in my opinion, it would be good to illustrate it in a diagram.
- All indexes in symbols in text and equations should be checked carefully.
- Literature should be checked if there are no newer items. Especially from the last 18 months.
Best regards
Author Response
Please refer to the attachment for detailed modifications. The manuscript has been polished at MDPI.

Reviewer 2 Report
This paper presents a segmentation model for ECU excitation signals. The model is described in detail, and it has been confirmed by the simulation results. In regard, the reviewer has the following concerns.
- Is 74% similarity high enough to replace the hardware-based simulation? 26% of the signals are different from the actual hardware, which may jeopardize the vehicle frequently.
- In Section 1, what does the 90%-reducible testing effort mean exactly? Money? Complexity? Else?
- At the end of Section 1, rather than explicitly showing the URL within the text, put it into the references.
- There must be more models other than the HBS-EED. However, only the HBS-EED is compared with the proposed model.
Author Response

(The authors gave the same response as above.)

Reviewer 3 Report
This paper proposes a method to simulate ECU excitation signals, specifically the crankshaft and camshaft position signals, with a piecewise function. The simulated signal is built with a combination of simple functions like sinewaves and square waves.
The paper is generally well written; however, I suggest some minor improvements:
- lines 154-155: at a first glance, reference [22] seems out of context, as it is a work related to signal processing for speech recognition. Further on into the manuscript, it appears that referenece [22] shares some similarities with this paper regarding the employed methodology. If this is the case, I suggest adding a sentence to explain better why reference [22] is relevant when first introducing it.
- line 235: it would be better to provide explicitely the measurement unit of the signal frequency (is it 1000Hz? 1000rad/s?)
- please correct the horizontal axis label in figures 5 and 18 ("frequence" should be changed to "frequency").
Author Response

(The authors gave the same response as above.)

Round 2
Reviewer 1 Report
Dear Authors,
Thank you very much for introducing changes that have improved the quality of the article. I have no more comments.
Best regards
Reviewer 2 Report
All the concerns have been addressed. Thank you.